# MIL-100(Fe)-Based Composite Films for Food Packaging

**DOI:** 10.3390/nano13111714

**Published:** 2023-05-23

**Authors:** Alexandra M. Pak, Elena A. Maiorova, Elizaveta D. Siaglova, Teimur M. Aliev, Elena N. Strukova, Aleksey V. Kireynov, Alexey A. Piryazev, Valentin V. Novikov

**Affiliations:** 1Nesmeyanov Institute of Organoelement Compounds, Russian Academy of Sciences, Vavilova Str. 28, 119991 Moscow, Russia; 2Moscow Institute of Physics and Technology, National Research University, Institutskiy per. 9, 141700 Dolgoprudny, Russia; 3Gause Institute of New Antibiotics, Russian Academy of Sciences, B. Pirogovskaya Str. 11/1, 119021 Moscow, Russia; 4Scientific and Educational Center “Composites of Russia”, Bauman Moscow State Technical University, 2nd Baumanskaya Str. 5, 105005 Moscow, Russia; 5Research Center for Genetics and Life Sciences, Scientific Direction Biomaterials, Sirius University of Science and Technology, 1 Olympic Ave, 354340 Sochi, Russia

**Keywords:** biocompatible metal–organic frameworks, active food packaging, hydrocolloids, composite materials

## Abstract

A biocompatible metal–organic framework MIL-100(Fe) loaded with the active compounds of tea tree essential oil was used to produce composite films based on κ-carrageenan and hydroxypropyl methylcellulose with the uniform distribution of the particles of this filler. The composite films featured great UV-blocking properties, good water vapor permeability, and modest antibacterial activity against both Gram-negative and Gram-positive bacteria. The use of metal–organic frameworks as containers of hydrophobic molecules of natural active compounds makes the composites made from naturally occurring hydrocolloids attractive materials for active packaging of food products.

## 1. Introduction

Materials for active food packaging are among the most trending research topics in the food industry [1]. Increasing the shelf life of food products [2] by inhibiting food spoilage through direct or indirect interactions with the active agents in the packaging, these composite materials are sought to solve global environmental problems, such as plastic and food waste [3]. The sustainability of their production from renewable resources, often waste [4], is crucial in the era of the depletion of natural resources, especially fossil fuels [5].

The sustainability and biocompatibility of such materials motivated the researchers to consider natural compounds both as matrices and active agents in active food packaging. The focus of this research is the materials produced from natural hydrocolloids, polysaccharide-, and protein-based water-soluble polymers able to form strong three-dimensional networks in aqueous solutions. They include starch [6], cellulose derivatives [7], chitosan [8], carrageenans [9], and others [10,11,12,13] that feature high biocompatibility [14], biodegradability [15], availability and variability [16], and good barrier properties [17]. However, the low affinity of hydrocolloids to hydrophobic molecules of most natural active compounds, such as plant essential oils (EOs) [18], hinders their industrial use in active food packaging.

Of the popular active agents in functional food packaging, EOs are complex mixtures of bioactive molecules that often act in synergy and are hard to isolate [19]. Tea tree essential oil (TTO) from *Melaleuca alternifolia*, which contains monoterpenes, sesquiterpenes, and their alcohol derivatives [20], is known for its antimicrobial and antioxidant properties that are rendered into active food packaging. Incorporation of TTO into the polymer matrix, however, is still a challenge because of the hydrophobicity and volatility of its components.

To overcome this limitation, the bioactive molecules of TTO may be encapsulated—by coating or entrapment—in a medium that protects them from degradation and volatilization and ensures their controlled release [21]. Nanoparticles (NPs) are good candidates for this purpose, as they provide a high specific surface area for active compounds to attach to [22].

Metal–organic frameworks (MOFs) [23] are a unique class of crystalline materials with exceptional porosity and tunability of their properties due to a smart choice of metal nodes and organic linkers [24]. Biocompatible MOFs are already used in catalysis [25], the separation of gases [26] and chiral compounds [27], drug delivery [28], and medicinal diagnostics [29]. Recently, they were also proposed as carriers of active compounds in food packaging with improved antibacterial performance [30] and controlled release of ethylene [31]. Bioactive components in some MOFs [32] make them potential active agents by themselves [33].

One of the most popular biocompatible MOFs is MIL-100(Fe), Fe_3_O(H_2_O)_2_OH(BTC)_2_ (H_3_BTC-benzene-1,3,5-tricarboxylic acid). Owing to its low cytotoxicity [34], remarkable hydrothermal stability [35], and very high specific surface area and porosity [36], it is targeted for biomedical applications [34]. To our knowledge, however, there is still a gap in the use of MIL-100(Fe) in food packaging.

Here, we attempted to close this gap by probing TTO-loaded MIL-100(Fe) NPs as an antimicrobial agent in composite films based on a mixture of kappa-carrageenan (Kc) and hydroxypropyl methylcellulose (HPMC).

## 2. Materials and Methods

### 2.1. Materials and Chemicals

Kappa-carrageenan (Kc, 0.3% aqueous solution viscosity at 25 °C: 5–25 cP), hydroxypropyl methylcellulose (HPMC, 2% aqueous solution viscosity at 20 °C: 40–60 cP), glycerol (≥99.5%), benzene-1,3,5-tricarboxylic acid (H_3_BTC, 95%), iron(II) chloride tetrahydrate (≥99.0%), sodium hydroxide (≥98%), and potassium sorbate (KS, 99%) were purchased from Sigma Aldrich (St. Louis, MO, USA). Tea tree essential oil (TTO) was purchased from Siberina LLC (Kirov, Russia). Mueller–Hinton (MH) broth and agar solid medium (Becton Dickinson and Company) were purchased from Hem Ltd. (Moscow, Russia)

### 2.2. Synthesis of MIL-100(Fe)

The microcrystalline MIL-100(Fe) powder was prepared as described previously [37]. An aqueous solution (25 mL) of NaOH (24 mmol) and H_3_BTC (8 mmol) was added dropwise to an aqueous solution (100 mL) of FeCl_2_·4H_2_O (12 mmol) under stirring, which immediately produced a green precipitate. The resulting solution was stirred at room temperature for 24 h until the precipitate changed its color from green to reddish brown. The solid was filtered and washed with hot ethanol and, several times, with hot distilled water. The sample was dried at room temperature.

### 2.3. Preparation of MIL-100(Fe)-TTO

MIL-100(Fe) was activated under vacuum at 120 °C for 24 h. The resulting dry powder (500 mg) was placed in a Schlenk frit heated to 85 °C and attached to the round bottom flask containing TTO (5 mL). Those were then attached to the vacuum distillation apparatus and the pressure was slowly reduced. The vacuum distillation of TTO through the layer of activated MIL-100(Fe) powder was performed at 85 °C until the liquid stopped collecting in the receiving flask. The resulting powder of MIL-100(Fe)-TTO was placed on a Petri dish and air-dried for 24 h.

### 2.4. Preparation of Composite Films

Composite films were fabricated using the Dr. Blade technique. Different amounts of MIL-100(Fe) or MIL-100(Fe)-TTO powders (0, 0.5, 1, 2, and 5 wt% of the total hydrocolloid weight) were dispersed under sonication in a distilled water solution (100 mL) of glycerol (0.8 g) and potassium sorbate (0.02 g) for 30 min. Kappa-carrageenan (1.6 g) and HPMC (0.4 g) were then slowly added under agitation. The mixture was heated to 70 °C and stirred for 30 min to produce an even suspension. The film-forming solution was degassed, spread on a clean silicon sheet heated to 45 °C using a blade (substrate–blade gap was set at 200 μm), and dried at 45 °C for 24 h. The resulting composite films—KcHPMC, KH/MIL_0.5–5_, and KH/MIL/TTO_0.5–5_ (Table 1)—were gently peeled off the silicon sheet and stored in a dark and dry place at ambient temperature prior to the measurements.

### 2.5. Scanning Electron Microscopy (SEM)

High-resolution surface images of the composite film samples were obtained using the Zeiss CrossBeam 550 Scanning Electron Microscope (Carl Zeiss Microscopy GmbH, Oberkochen, Germany). Images were collected using an in-lens detector operating at 5 kV.

### 2.6. Thickness

The thickness of the composite film samples was measured in 10 random places using a digital micrometer. The mean values were also used to probe the mechanical and optical properties of the films.

### 2.7. Mechanical Properties

Mechanical properties, such as tensile strength, elastic modulus, and elongation at break, were measured for the composite film samples (750 mm long and 13 mm wide) prepared in accordance with ASTM D882 [38]. Measurements were conducted using a universal testing machine (Zwick/Roell Z2.5, Zwick GmbH & Co., Ulm, Germany) operating with a 50 N load cell with a traction speed of 50 mm/min. All assays were performed at 30 ± 2 °C with a relative humidity of 50 ± 2%. Each sample was tested at least 5 times in parallel. The elastic modulus was obtained as the slope of an elastic (linear) region of the stress/strain curves. The tensile strength (TS) (Equation (1)) and the elongation at break (EAB) (Equation (2)) were obtained as follows:(1)$$TS\left(MPa\right)=\frac{F}{S},$$
(2)$$EAB\left(\%\right)=\frac{∆L}{L_{0}}\times 100\%,$$
where F is the breaking force exerted (N), S is the cross-sectional film area (mm^2^), ∆L is the film elongation at the end of the procedure (mm), and L_0_ is the initial film elongation (mm).

### 2.8. Water Vapor Permeability (WVP)

The water vapor permeability of the composite film samples was evaluated using the gravimetric method in accordance with ASTM E96 adapted to hydrophilic films [39]. The film samples were cut into circles and fixed in a glass vial containing 6 g of dry silica gel (0% RH) with the distance between the silica gel and the composite film about 2 cm. The vials were kept in a desiccator containing a saturated solution of NaCl with an excess of undissolved NaCl at 25 ± 2 °C and 75% RH for 2 h. After the initial incubation period, a steady state had been achieved and periodic weighing of vials without film samples provided the amount of water permeated through the film for 24 h. WVP was defined using Equation (3):(3)$$WVP=\frac{A\times ∆x}{∆p\times S,}$$
where WVP is the water vapor permeability (g·mm/Pa·h·m^2^), A (g/h) is the slope of linear regression of the vial weight vs. the time graph, Δx (mm) is the thickness of the film sample, Δp (2.377 kPa) is the difference in partial pressure, and S (m^2^) is the exposed surface area of the film. Each experiment was repeated at least three times.

### 2.9. X-ray Powder Diffraction

Powder X-ray diffraction (PXRD) data were collected on a PROTO AXRD benchtop instrument equipped with a Dectris Mythen 1K 1D-detector, using nickel-filtered CuKα (λ = 0.154056 Å) radiation. The scan was performed in a 2θ range of 5–25° with a 0.02° step.

### 2.10. Light Transmittance and Transparency

Light transmittance of the composite film samples was measured at the ultraviolet and visible ranges (220–500 nm) using a UV-vis spectrophotometer UV-2600 (Shimadzu, Kyoto, Japan). The transparency value (TV) of the film was calculated using Equation (4):(4)$$TV=\frac{-\log T_{500}}{∆x},$$
where T_500_ is the fractional transmittance at 500 nm and Δx (mm) is the film thickness. The higher transparency value corresponds to the lower transparency of the film.

### 2.11. Indentification of the Components of TTO in MIL-100(Fe)-TTO

MIL-100(Fe)-TTO powder (10 mg) was placed in a 1M HCl solution (2 mL) and diluted with distilled water (8 mL). After the complete hydrolysis of MOF, the aqueous phase was separated from the precipitate and extracted with methylene chloride (10 mL); the organic fraction was separated and analyzed using a GCMS-QP2020 (Shimadzu, Kyoto, Japan) with a quadrupole detector. Analysis conditions: sample volume—1 µL; injector temperature −250 °C; column SH-RTx-5MS (30 m, 0.25 mm, 0.25 μm); thermostat temperature—40 °C, 1 min -> heating to 290 °C, 30°C/min -> 290 °C, 2 min; carrier gas-helium, column flow rate—1 mL/min; ion source temperature—200 °C; and interface temperature—250 °C. The contents of the TTO components in the test sample were estimated by comparing their retention time (min), peak area, peak height, and mass spectra patterns with those from the database of authentic compounds stored in the library of the National Institute of Standards and Technology (NIST).

### 2.12. Antibacterial Properties of Composite Films

The antibacterial activity of different composite films samples was probed against Gram-negative bacteria, *Escherichia coli* (*E. coli* ATCC 25922), and Gram-positive bacteria, *Staphylococcus aureus* (*S. aureus* ATCC 6538p), using the standard colony counting method [30]. Fresh bacterial cultures in the Mueller–Hinton (MH) broth were obtained via incubation at 37 °C. Bacterial suspensions were then adjusted to ~10^7^ CFU/mL with sterilized 9% NaCl solution. Different samples were cut into circles with a diameter of 5 cm and immersed into bacterial suspensions (30 mL) in the culture dish. The supernatant was taken out at 1, 3, and 5 h after the addition of the samples and it was diluted to a proper concentration with a 9% NaCl solution. The resulting solutions (100 μL) were then spread on the surface of MH agar solid media. The solid media were cultured at 37 °C for 18 h, and the viable numbers of *E. coli* and *S. aureus* colonies were counted via visual observation. The *E. coli* and *S. aureus* suspension without any samples was cultured on an MH agar solid medium as a control. Each experiment was repeated at least three times.

### 2.13. Statistical Analysis

Statistical analysis was performed using DSAASTAT, ver. 1.541, by Andrea Onofri (Perugia, Italy). The data were analyzed using the analysis of variance (ANOVA), and multiple comparisons of the means were made using the Tukey’s test (*p* < 0.05).

## 3. Results and Discussion

### 3.1. Characterizations of MIL-100(Fe) and MIL-100(Fe)-TTO

MIL-100(Fe) was synthesized via the sustainable protocol in water at room temperature as described previously [37] to produce a precipitate with high crystallinity in the absence of any inorganic corrosive acid, such as HF and HNO_3_ that are conventionally used to induce crystal growth [40]. The use of the iron(II) salt prevents the formation of semi-amorphous Fe-BTC (a MOF material, commercialized as Basolite F300, with the same metal ions and organic linkers) that occurs under the same conditions from an iron(III) salt. Although Fe-BTC can be reconstructed in water for high crystallinity using the recently emerged approach [41], the process is time-consuming. The formation of the target MIL-100(Fe) under the chosen conditions was confirmed using powder X-ray diffraction (Figure 1a). In agreement with the data reported earlier [40], the XRD pattern featured both groups of the characteristic peaks of MIL-100(Fe) at 5–7° and 10–11° 2θ ranges.

TTO was encapsulated into MIL-100(Fe), which was activated under vacuum at 120 °C for 24 h to remove all the solvent molecules from the pores, using vacuum distillation through the layer of powdered MIL-100(Fe), as this technique ensures better adsorption of TTO as compared to the conventional post-synthetic incorporation via soaking. The XRD pattern of the powder sample of the resulting MIL-100(Fe)-TTO (Figure 1b) showed no changes in the position of the diffraction peaks upon the encapsulation of TTO; however, the change in relative intensities of both groups of the characteristic peaks (an increase for the group at 5–7° 2θ and a decrease for the group at 10–11° 2θ) might hint some minor changes in the crystal structure of MIL-100(Fe) upon the encapsulation of the chosen guest.

The components of TTO in MIL-100(Fe)-TTO were identified by analyzing the composition of the organic matter remaining from acid hydrolysis using GC-MS (Figure 2). The comparison to the sample of TTO prepared under the same conditions showed that all the main components of TTO, except for terpinen-4-ol, were present in MIL-100(Fe)-TTO (Table 2). A higher content of p-cymene in the latter may result from stacking interactions of its aromatic ring with BTC^2-^ linkers that facilitate its encapsulation in the pores of MIL-100(Fe).

### 3.2. Characterizations of Composite Films

The composite films containing MIL-100(Fe) or MIL-100(Fe)-TTO as a filler at different concentrations (0.5, 1, 2, and 5 wt% of the total hydrocolloid weight) and a mixture of Kc and HPMC in a 4:1 ratio as a polymer matrix were fabricated using the Dr. Blade technique via a home-built coating machine to produce a layer of the film-forming solution with the same thickness for all compositions. The MIL-100(Fe) and MIL-100(Fe)-TTO powders were ultrasonically pre-treated to evenly disperse the particles of these MOFs in the film-forming solution before the addition of hydrocolloids, as the increase in the viscosity of the solution caused clumping of the powders and an uneven distribution of the filler particles. A 2 mm substrate–blade gap was chosen as the biggest height that the film-forming solution was able to retain at 45 °C.

The resulting composite films containing MIL-100(Fe) and MIL-100(Fe)-TTO featured good transparency with the brown color of MIL-100(Fe) becoming more intense with the increase in the content of the filler (Figure 3a,c). The particles of MIL-100(Fe) or MIL-100(Fe)-TTO (50–400 nm) were uniformly distributed in the films (Figure 3b,d) which kept the microstructure of the matrix at lower concentrations of these filles. At concentrations above 2%, however, the aggregation of the particles (0.5–5 μm) resulted in a rougher surface on the composite films.

The incorporation of MIL-100(Fe) or MIL-100(Fe)-TTO into the composite films was confirmed using powder X-ray diffraction (Figure 1). The latter showed the presence of the characteristic peaks of MIL-100(Fe) at 10–11° 2θ in the XRD patterns of all the samples and a gradual increase in their intensities with an increase in the content of the filler in the continuous phase of the amorphous matrix.

Transparency has a great impact on the appearance of a packaged product. Therefore, UV-vis spectroscopy was used to assess the light transmittance of the obtained films (Figure 4). The control film that contained no filler featured the lowest transmittance of 2.07 at 500 nm (Table 3), which is the highest transparency of the films. However, it dramatically increased upon the incorporation of MIL-100(Fe) or MIL-100(Fe)-TTO following the above color change with an increase in the content of these fillers. The difference between the composite films containing MIL-100(Fe) and MIL-100(Fe)-TTO may arise from the higher mass and, therefore, a lower amount of the embedded MOF particles when loaded with TTO. The films block the UV light up to 320 nm, which can be useful for reducing its harmful impact on foodstuff during long-term storage.

Good mechanical properties are crucial for the applications of composite materials in food packaging. While the film thickness gradually increased with the content of MIL-100(Fe) and MIL-100(Fe)-TTO, the presence of 0.5–1% of these fillers in the composition caused the decrease in the tensile strength, elongation at break, and elastic modulus of the composite films (Table 3). At higher contents of MIL-100(Fe) and MIL-100(Fe)-TTO (to 2% and 5%, respectively), however, these mechanical properties improved, possibly by the compaction of the polymer matrix that occurs along with the aggregation of the filler particles. The tensile strength of the films is comparable to that of high-density polyethylene (HDPE, 15–45 MPa) and low-density polyethylene (LDPE, 7–14 MPa) commonly used in commercial packaging [42].

Water vapor permeability (WVP), which describes the moisture migration between food components and the surrounding atmosphere, is another important characteristic of food packaging. The composite films containing MIL-100(Fe) and MIL-100(Fe)-TTO show different behaviors (Table 3). The addition of MIL-100(Fe)-TTO to the composition resulted in an increase in WVP that correlated with the content of this filler. In contrast, the films with MIL-100(Fe) had better moisture barrier properties at lower concentrations (0.5%). This may be explained by bonding between the sulfate groups of κ-carrageenan and the metal ions on the surface of MIL-100(Fe) NPs that is inhibited by aggregation at higher concentrations (1–5%) and in the presence of a hydrophobic layer of TTO on MIL-100(Fe)-TTO NPs.

The antibacterial properties of the films were tested against *E. coli* and *S. aureus* using the standard colony counting method (Figure 5). The films based on MIL-100(Fe) emerged as more active against both pathogens than those based on MIL-100(Fe)-TTO (Table 4 and Table 5). A plausible reason behind the lower antibacterial activity of the latter is the release of the iron(III) ions and BTC^2−^ anions upon the decomposition of MIL-100(Fe) in a saline environment inhibited by a hydrophobic layer of TTO on the surface of NPs. The hydrophobicity of the components of TTO and its low concentration in MIL-100(Fe)-TTO may also be contributed to the rather poor performance of the composite films based on MIL-100(Fe)-TTO.

## 4. Conclusions

The use MIL-100(Fe) loaded with the active compounds of tea tree essential oil as a filler for the hydrocolloid-based composite films improved their UV-blocking ability without a deterioration in the mechanical properties. The latter were comparable—at the highest concentration of the filler—to those of commercial LDPE and HDPE, which cannot be achieved upon the direct incorporation of essential oils into the hydrocolloid matrix, as it produces brittle materials based on κ-carrageenan [43,44] and hydroxypropyl methylcellulose [45,46]. The films with MIL-100(Fe) also showed modest antibacterial activity against both Gram-negative (*Escherichia coli*) and Gram-positive (*Staphylococcus aureus*) bacteria while the films with MIL-100(Fe)-TTO did not have any significant bactericidal effect on these foodborne pathogens. These composite materials, therefore, may emerge as active packaging or active coating on foods. Further studies are, however, needed to assess their effect on the packaged products; they are underway within our group.

## Figures and Tables

**Figure 1 nanomaterials-13-01714-f001:**
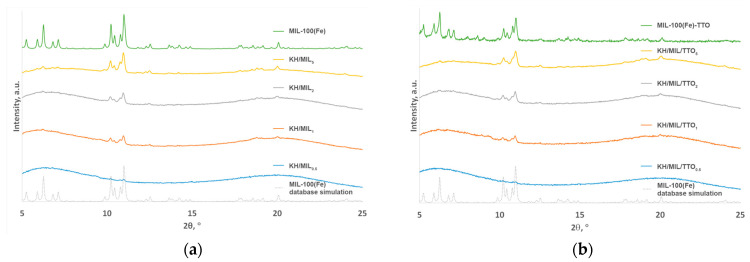
XRD patterns of powdered MIL-100(Fe) (**a**) and MIL-100(Fe)-TTO (**b**), of the composite films containing MIL-100(Fe) 0.5, 1, 2, and 5 wt% of the total hydrocolloid weight) (**a**) and MIL-100(Fe)-TTO (0.5, 1, 2, and 5 wt% of the total hydrocolloid weight) (**b**), and of pure MIL-100(Fe) from the Cambridge Crystallographic Database, Refcode CIGXIA [41].

**Figure 2 nanomaterials-13-01714-f002:**
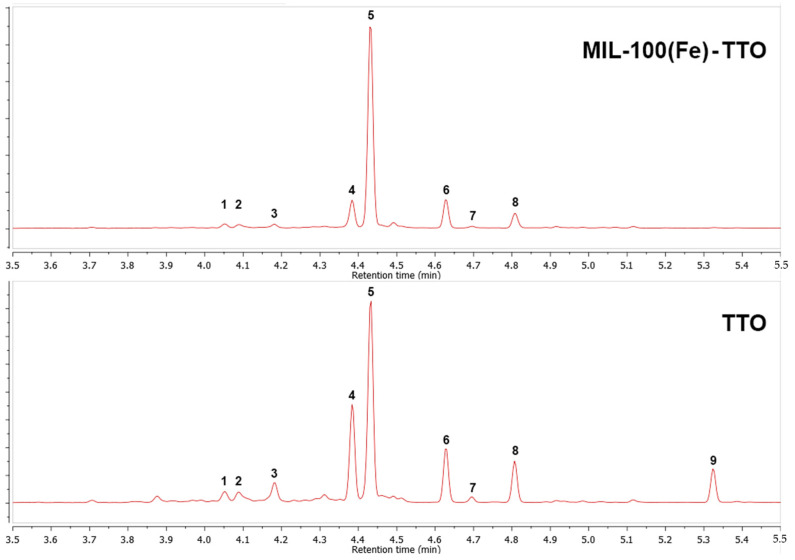
GC chromatograms of MIL-100(Fe)-TTO powder and a TTO sample. The numbered peaks are assigned in Table 2.

**Figure 3 nanomaterials-13-01714-f003:**
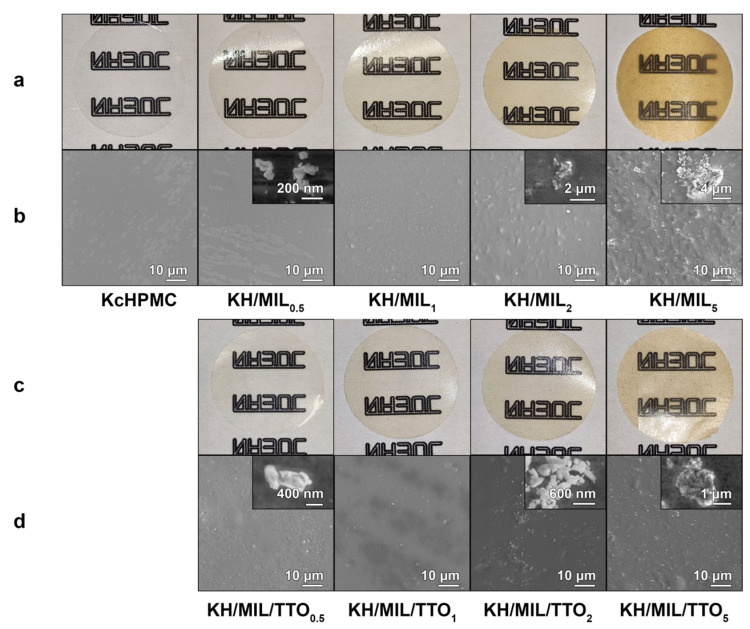
Digital photos (**a**,**c**) and SEM images of the surface (**b**,**d**) of the composite films without the filler and with its different concentrations (0.5, 1, 2, and 5 wt% of the total hydrocolloid weight).

**Figure 4 nanomaterials-13-01714-f004:**
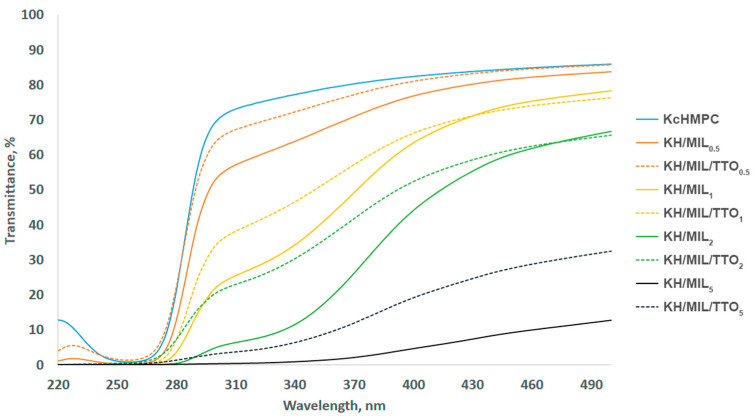
Light transmittance of the composite films without the filler and with its different concentrations (0.5, 1, 2, 5 wt% of the total hydrocolloid weight).

**Figure 5 nanomaterials-13-01714-f005:**
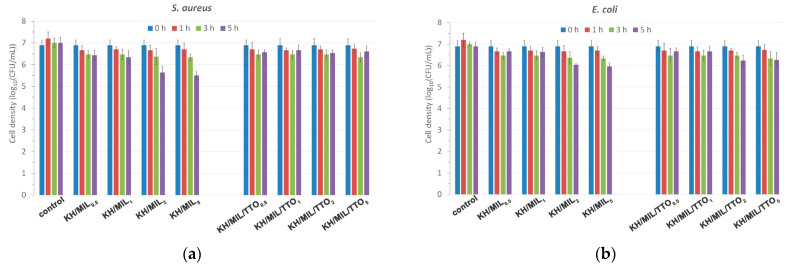
Antibacterial activity of the films with the different concentrations of the filler (0.5, 1, 2, and 5 wt% of the total hydrocolloid weight): (**a**) *S. aureus*; (**b**) *E. coli*.

**Table 1 nanomaterials-13-01714-t001:** Composition of film-forming solutions for the composite films.

Sample	Kc (g)	HMPC (g)	Glycerol (g)	KS (g)	MIL-100(Fe) (g)	MIL-100(Fe)-TTO (g)
KcHPMC	1.6	0.4	0.8	0.02	0	0
KH/MIL_0.5_	1.6	0.4	0.8	0.02	0.01	0
KH/MIL_1_	1.6	0.4	0.8	0.02	0.02	0
KH/MIL_2_	1.6	0.4	0.8	0.02	0.04	0
KH/MIL_5_	1.6	0.4	0.8	0.02	0.1	0
KH/MIL/TTO_0.5_	1.6	0.4	0.8	0.02	0	0.01
KH/MIL/TTO_1_	1.6	0.4	0.8	0.02	0	0.02
KH/MIL/TTO_2_	1.6	0.4	0.8	0.02	0	0.04
KH/MIL/TTO_5_	1.6	0.4	0.8	0.02	0	0.1

**Table 2 nanomaterials-13-01714-t002:** Compounds identified in MIL-100(Fe)-TTO powder and a TTO sample.

Peak	Component	Spectral Similarity (%)
1	1,3-Cyclopentadiene, 5,5-dimethyl-1-propyl-	89
2	Camphene	89
3	3-p-Menthene	96
4	4-Carene	96
5	p-Cymene	96
6	γ-Terpinene	96
7	p-Mentha-3,8-diene	95
8	2-Carene	94
9	Terpinen-4-ol	97

**Table 3 nanomaterials-13-01714-t003:** Thickness, mechanical (tensile strength, TS; elongation at break, EAB; elastic modulus, EM), barrier (water vapor permeability, WVP), and optical (transmittance value, TV) properties of the composite films without the filler and with its different concentrations (0.5, 1, 2, and 5 wt% of the total hydrocolloid weight).

Sample	Thickness (μm)	TS (MPa) ^1^	EAB (%)	EM (GPa)	WVP (g·mm/Pa·h·m^2^) ^1^	TV ^1^
KcHPMC	29 ± 3 ^a^	59.30 ± 9.69 ^a^	5.6 ± 1.1 ^ab^	2.8 ± 0.6 ^a^	0.373 ± 0.031 ^a^	2.07 ^a^
KH/MIL_0.5_	32 ± 2 ^b^	55.66 ± 4.26 ^ab^	3.7 ± 0.1 ^ab^	2.6 ± 0.2 ^ab^	0.331 ± 0.024 ^a^	2.40 ^a^
KH/MIL_1_	39 ± 1 ^c^	46.27 ± 2.97 ^abc^	6.0 ± 0.7 ^a^	2.2 ± 0.1 ^abc^	0.380 ± 0.034 ^ab^	2.73 ^a^
KH/MIL_2_	40 ± 2 ^c^	53.65 ± 5.72 ^ab^	6.0 ± 0.5 ^a^	2.5 ± 0.2 ^ab^	0.468 ± 0.032 ^cd^	4.41 ^b^
KH/MIL_5_	48 ± 3 ^d^	36.88 ± 2.48 ^c^	3.2 ± 0.6 ^b^	1.7 ± 0.1 ^c^	0.585 ± 0.029 ^e^	18.56 ^d^
KH/MIL/TTO_0.5_	33 ± 2 ^b^	52.05 ± 2.91 ^ab^	4.0 ± 0.8 ^ab^	2.6 ± 0.1 ^ab^	0.451 ± 0.033 ^bc^	2.14 ^a^
KH/MIL/TTO_1_	41 ± 1 ^cd^	43.59 ± 2.50 ^bc^	3.8 ± 0.3 ^ab^	2.5 ± 0.2 ^ab^	0.508 ± 0.028 ^cde^	2.85 ^a^
KH/MIL/TTO_2_	39 ± 3 ^c^	35.22 ± 3.40 ^c^	3.9 ± 0.7 ^ab^	2.0 ± 0.1 ^bc^	0.511 ± 0.035 ^cde^	4.70 ^b^
KH/MIL/TTO_5_	43 ± 2 ^d^	55.70 ± 2.69 ^a^	5.0 ± 0.6 ^ab^	2.7 ± 0.2 ^a^	0.540 ± 0.041 ^de^	11.35 ^c^

^1^ Properties are normalized to the thickness of the samples. ^a–e^ Different superscripts within the same column indicate significant differences among the formulations (*p* < 0.05).

**Table 4 nanomaterials-13-01714-t004:** Antibacterial activity of the films with the different concentrations of the filler (0.5, 1, 2, and 5 wt% of the total hydrocolloid weight) against *S. aureus*.

Sample	Cell Density (lg(CFU/mL))			
	0 h	1 h	3 h	5 h
Control	6.8 ± 0.2 ^c/d^	7.3 ± 0.3 ^a/d^	7.2 ± 0.2 ^ab/d^	7.0 ± 0.3 ^bc/d^
KH/MIL_0.5_	6.8 ± 0.2 ^a/d^	6.8 ± 0.2 ^a/de^	6.3 ± 0.2 ^b/efg^	6.4 ± 0.2 ^c/de^
KH/MIL_1_	6.8 ± 0.2 ^a/d^	6.8 ± 0.1 ^ab/de^	6.1 ± 0.3 ^b/efg^	6.3 ± 0.3 ^ab/e^
KH/MIL_2_	6.8 ± 0.2 ^a/d^	6.8 ± 0.2 ^a/de^	5.9 ± 0.4 ^b/fg^	5.6 ± 0.3 ^b/f^
KH/MIL_5_	6.8 ± 0.2 ^a/d^	6.4 ± 0.3 ^ab/e^	5.8 ± 0.2 ^bc/g^	5.5 ± 0.2 ^c/f^
KH/MIL/TTO_0.5_	6.8 ± 0.2 ^a/d^	6.8 ± 0.3 ^a/de^	6.6 ± 0.2 ^a/de^	6.6 ± 0.1 ^a/de^
KH/MIL/TTO_1_	6.8 ± 0.2 ^a/d^	6.8 ± 0.1 ^a/de^	6.6 ± 0.2 ^a/de^	6.7 ± 0.2 ^a/de^
KH/MIL/TTO_2_	6.8 ± 0.2 ^a/d^	6.8 ± 0.2 ^a/de^	6.5 ± 0.2 ^a/def^	6.5 ± 0.2 ^a/de^
KH/MIL/TTO_5_	6.8 ± 0.2 ^a/d^	6.8 ± 0.2 ^a/de^	6.5 ± 0.2 ^ab/def^	6.6 ± 0.3 ^b/de^

^a–c, d–g^ Different superscripts within the same row/column indicate significant differences among the formulations (*p* < 0.05).

**Table 5 nanomaterials-13-01714-t005:** Antibacterial activity of the films with the different concentrations of the filler (0.5, 1, 2, and 5 wt% of the total hydrocolloid weight) against *E. coli*.

Sample	Cell Density (lg(CFU/mL))			
	0 h	1 h	3 h	5 h
Control	6.9 ± 0.3 ^a/d^	7.2 ± 0.3 ^a/d^	7.0 ± 0.1 ^a/d^	6.9 ± 0.2 ^a/d^
KH/MIL_0.5_	6.9 ± 0.3 ^a/d^	6.7 ± 0.2 ^ab/d^	6.5 ± 0.2 ^b/d^	6.7 ± 0.1 ^ab/e^
KH/MIL_1_	6.9 ± 0.3 ^a/d^	6.7 ± 0.2 ^a/d^	6.5 ± 0.2 ^a/d^	6.6 ± 0.2 ^a/e^
KH/MIL_2_	6.9 ± 0.3 ^a/d^	6.7 ± 0.3 ^ab/d^	6.4 ± 0.3 ^ab/d^	6.0 ± 0.1 ^b/f^
KH/MIL_5_	6.9 ± 0.3 ^a/d^	6.7 ± 0.2 ^a/d^	6.3 ± 0.1 ^b/d^	6.0 ± 0.2 ^c/f^
KH/MIL/TTO_0.5_	6.9 ± 0.3 ^a/d^	6.7 ± 0.4 ^a/d^	6.5 ± 0.3 ^a/d^	6.7 ± 0.2 ^a/e^
KH/MIL/TTO_1_	6.9 ± 0.3 ^a/d^	6.7 ± 0.2 ^ab/d^	6.5 ± 0.3 ^b/d^	6.7 ± 0.2 ^ab/e^
KH/MIL/TTO_2_	6.9 ± 0.3 ^a/d^	6.7 ± 0.1 ^a/d^	6.5 ± 0.2 ^a/d^	6.2 ± 0.3 ^a/ef^
KH/MIL/TTO_5_	6.9 ± 0.3 ^a/d^	6.7 ± 0.3 ^a/d^	6.3 ± 0.3 ^b/d^	6.4 ± 0.2 ^ab/ef^

^a–c, d–f^ Different superscripts within the same row/column indicate significant differences among the formulations (*p* < 0.05).

## Data Availability

Data presented in this study are available on request from the corresponding author.

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
