# Peer review of "MIL-100(Fe)-Based Composite Films for Food Packaging"

_nanomaterials, 2023, doi:10.3390/nano13111714_

Round 1
Reviewer 1 Report
The manuscript entitled “MIL-100(Fe)-based composite films for food packaging”, submitted for evaluation to Nanomaterials, presents the proposal of complex biomaterial designed for food packaging, as an alternative for hardly biodegradable e.g. polyethylene foils.. In the context of the overwhelming wastes problem, I consider the topic of the work to be important and timely.
In general, the work clear and written in the correct language. The structure of the manuscript is clear. However, I see some aspects which should be discussed and supplemented. My comments and questions concerning the submitted article are listed below:
COMMENTS TO AUTHORS
1. Sections 2.1. and 2.2.: “The microcrystalline MIL-100(Fe) powder was prepared as described previously [36].
2.2. Synthesis of MIL-100(Fe).
Aqueous solution (25 mL) of NaOH (24 mmol) and H3BTC (8 mmol) was added dropwise to aqueous solution (100 mL) of FeCl2·4H2O (12 mmol) under stirring. The resulting solution was stirred at room temperature for 24 h until the precipitate changed its color from green to reddish brown. The solid was filtered and washed with hot ethanol and, several times, with hot distilled water. The sample was dried at room temperature.”
The last sentence of section 2.1. and entire section 2.2. deals with the same process and should be combined.
2. Composite biofilm sample names are confusing as they are too similar to MIL-100(Fe)-TTO powder sample designation. I suggest you consider changing the names of composite film samples.
3. Section 2.13.: Please provide the medium name (and solid or liquid one?) for bacterial culture preparation. Names of bacteria should be in italics. Supernatant collection requires earlier centrifugation – has it been performed? Line 168-169: E. coli CFU were counted but how about S. aureus?
4. Line 204: you mention “Control film without the filler featured the highest light transmittance of 85.9% (Table 1),”. However, table 1 did not shopw the values of light transmittance intensity. It there any table missing?
5. Lines 228-229: “In contrast, the films with MIL-100(Fe) had better moisture barrier properties at lower concentrations (0.5%).”. I do not see any contrast, the tendencies are similar for both series of films.
6. Antibacterial activity: the conclusion that saples show antibacterial activity is risky because of no SD or statistical data were calculated and presented, especially that reduction of cell density is not significant. Statistical analysis should be performed and added.
7. Conclusions: “The use MIL-100(Fe) loaded with the active compounds of tea tree essential oil as a filler for the hydrocolloid-based films improved their UV-blocking ability without a deterioration of the mechanical properties.” . I think that this conclusion cannot be withdrawn because it seems from the data that mecahnical parameters are dependent on KcHPMC rather than on addition of MIL-100(Fe) (please see Table 2).
Author Response
Comment 1:
Sections 2.1. and 2.2.: “The microcrystalline MIL-100(Fe) powder was prepared as described previously [36].
2.2. Synthesis of MIL-100(Fe).
Aqueous solution (25 mL) of NaOH (24 mmol) and H3BTC (8 mmol) was added dropwise to aqueous solution (100 mL) of FeCl2·4H2O (12 mmol) under stirring. The resulting solution was stirred at room temperature for 24 h until the precipitate changed its color from green to reddish brown. The solid was filtered and washed with hot ethanol and, several times, with hot distilled water. The sample was dried at room temperature.”
The last sentence of section 2.1. and entire section 2.2. deals with the same process and should be combined.
Author’s reply: We modified Sections 2.1 and 2.2 as instructed by Reviewer.
Comment 2: Composite biofilm sample names are confusing as they are too similar to MIL-100(Fe)-TTO powder sample designation. I suggest you consider changing the names of composite film samples.
Author’s reply: We changed the sample names for biofilms, as instructed. We also provided a new Table 1 with the film compositions to address Comment 3 of the Reviewer 3 (see below).
Comment 3: Section 2.13.: Please provide the medium name (and solid or liquid one?) for bacterial culture preparation. Names of bacteria should be in italics. Supernatant collection requires earlier centrifugation – has it been performed? Line 168-169: E. coli CFU were counted but how about S. aureus?
Author’s reply: We revised the description of the cell experiments as instructed by Reviewer. No additional centrifugation was performed prior to the supernatant collection to allow for a comparison with other results from our group that used this protocol.
Comment 4: Line 204: you mention “Control film without the filler featured the highest light transmittance of 85.9% (Table 1),”. However, table 1 did not show the values of light transmittance intensity. It there any table missing?
Author’s reply: These data now appear in Table 3.
Comment 5: Lines 228-229: “In contrast, the films with MIL-100(Fe) had better moisture barrier properties at lower concentrations (0.5%).”. I do not see any contrast, the tendencies are similar for both series of films.
Author’s reply: The general trend in the moisture barrier properties is indeed very similar for both series of the films; however, at the smallest MOF concentration (0.5% ) MIL-100(Fe) has a lower value of WVP (0.331) as compared to a control film (0.373) or a TTO-containing film (0.468). These values now appear in Table 3.
Comment 6: Antibacterial activity: the conclusion that samples show antibacterial activity is risky because of no SD or statistical data were calculated and presented, especially that reduction of cell density is not significant. Statistical analysis should be performed and added.
Author’s reply: We performed the additional statistical analysis to address this concern raised by Reviewer. Its results now appear in Fig.5 as standard deviation bars.
Comment 7: Conclusions: “The use MIL-100(Fe) loaded with the active compounds of tea tree essential oil as a filler for the hydrocolloid-based films improved their UV-blocking ability without a deterioration of the mechanical properties.”. I think that this conclusion cannot be withdrawn because it seems from the data that mechanical parameters are dependent on KcHPMC rather than on addition of MIL-100(Fe) (please see Table 2).
Author’s reply: We do agree that the composition of the polymer matrix is the major factor that influences the mechanical properties of the composite films; however, we kept the composition of the hydrocolloid matrix constant across all the films. Therefore, the observed changes in mechanical properties can only be induced by the filler compound, i.e. MIL-100(Fe).

Reviewer 2 Report
The authors have prepared composite films by MIL-100(Fe) loaded tea tree essential oil based on к-carrageenan and hydroxypropyl methylcellulose with the uniform distribution of the particles of this filler. The authors have demonstrated that the films have superior UV-blocking properties, good water vapor permeability and modest antibacterial activity against both Gram-negative and Gram-positive bacteria. Overall, this work can inspire more material design ideas of metal-organic frameworks as containers for active packaging of food products. Therefore, I would like to recommend this work to publish in Nanomaterials. Below are some comments for the authors.
1. For the XRD patterns of the powders MIL-100(Fe) and MIL-100(Fe)-TTO of Figure 1, the characteristic peaks should be indicated and described in the main text.
2. The authors have indicated “The particles of MIL-100(Fe) or MIL-100(Fe)-TTO (50-150 nm) were uniformly distributed in the films (Fig. 3 b, d) that kept the micro-structure of the matrix at lower concentrations of these filles”. This paper would be more impressive if the authors could provide energy dispersive X-ray (EDX) mapping analysis of these films to reveal the distribution of MIL-100(Fe) or MIL-100(Fe)-TTO.
3. For the introduction “Nanoparticles (NPs) are good candidates for this purpose, as they provide high specific surface area for active compounds to attach to”, more references could be cited to broaden the introduction.
https://doi.org/10.2147/IJN.S328767
Author Response
Comment 1: For the XRD patterns of the powders MIL-100(Fe) and MIL-100(Fe)-TTO of Figure 1, the characteristic peaks should be indicated and described in the main text.
Author’s reply: The XRD patterns of these samples are discussed in the revised version of our ms. as instructed by Reviewer.
Comment 2: The authors have indicated “The particles of MIL-100(Fe) or MIL-100(Fe)-TTO (50-150 nm) were uniformly distributed in the films (Fig. 3 b, d) that kept the micro-structure of the matrix at lower concentrations of these filles”. This paper would be more impressive if the authors could provide energy dispersive X-ray (EDX) mapping analysis of these films to reveal the distribution of MIL-100(Fe) or MIL-100(Fe)-TTO.
Author’s reply: Unfortunately, the SEM employed did not have the operational EDX module thereby preventing us from performing the EDX analysis. However, the distribution of MIL-100(Fe) or MIL-100(Fe)-TTO can be easily assessed on the images on Fig. 3.
Comment 3: For the introduction “Nanoparticles (NPs) are good candidates for this purpose, as they provide high specific surface area for active compounds to attach to”, more references could be cited to broaden the introduction.
https://doi.org/10.2147/IJN.S328767
Author’s reply: We are grateful to Reviewer for providing this useful citation, which is now mentioned in the Introduction section of our ms.

Reviewer 3 Report
The paper "MIL-100(Fe)-based composite films for food packaging" by Alexandra M. Pak et al. reports the use of biocompatible metal-organic framework MIL-100(Fe) loaded with active compounds of tea tree essential oil producing composite films with an uniform distribution of the particles of this filler. The work is interesting, anyway I have some major concerns that should be considered:
In the materials and methods section, statistical analysis is missing, please add.
The acronym for H3BTC should be expanded: 2-Bromo-1,3,5 benzenetricarboxylic acid and included in material section.
Line 95: it would be useful to provide a table with sample name and compositions.
In figure 1: please specify the composition of this sample. Xray for all samples should be reported
In figure 3: For a better comparison among sample surfaces, SEM images should be provided at the same magnification.
In table2: sample thickness should be kept consistent among samples otherwise properties (such as WTR, UV transmittance and mechanical properties) will be affected accordingly and a comparison among samples is not possible.
In figure 5: standard deviation bars and blank sample as a control are missing. Please add both.
Line 249-251: this must be demonstrated with samples having the same thickness.
This conclusion is not clear. After a careful statistical analysis, it must be clearly stated that MIL-100(Fe) show superior antibacterial activity compared to the corresponding samples containing TTO that show none (or little) activity against staphylococcus a.
Author Response
Comment 1: In the materials and methods section, statistical analysis is missing, please add.
Author’s reply: We added an appropriate section (2.14) to “Materials and Methods”.
Comment 2: The acronym for H3BTC should be expanded: 2-Bromo-1,3,5 benzenetricarboxylic acid and included in material section.
Author’s reply: In the revised version of our ms., we explained (in the Introduction and the Section 2.1 of Materials and Methods) what this acronym stands for as instructed by Reviewer.
Comment 3: Line 95: it would be useful to provide a table with sample name and compositions.
Author’s reply: We also provided a new Table 1 with the film compositions as instructed by Reviewer.
Comment 4: In figure 1: please specify the composition of this sample. X-ray for all samples should be reported
Author’s reply: We added the XRD patterns for all the composites containing MIL-100 to Figure 1. The composition of the samples is now described in new Table 1.
Comment 5: In figure 3: For a better comparison among sample surfaces, SEM images should be provided at the same magnification.
Author’s reply: We are grateful to Reviewer for raising this issue, as it helped us to pinpoint a mistake in scale designation during the preparation of the SEM images, which now feature the correct particle sizes. Wide-shot SEM images were replaced to provide a better comparison of the samples with different compositions.
Comment 6: In table2: sample thickness should be kept consistent among samples otherwise properties (such as WTR, UV transmittance and mechanical properties) will be affected accordingly and a comparison among samples is not possible.
Author’s reply: We do agree that the absolute values of the obtained parameters for the films with different thicknesses cannot be compared to each other. However, the film preparation technique used in this study is based on slow evaporation of a solvent from the film-forming solutions with a constant volume, so the thickness of the final film cannot be controlled. To account for this drawback, we provided the values that were normalized to the thickness of the samples, as is the standard practice in this field. An appropriate comment was added to Table 3.
Comment 7: In figure 5: standard deviation bars and blank sample as a control are missing. Please add both.
Author’s reply: Fig. 5 was modified as instructed by Reviewer.
Comment 8: Line 249-251: this must be demonstrated with samples having the same thickness.
Author’s reply: As explained above, the film preparation technique does not allow controlling the thickness of the final film for different initial compositions. To account for this drawback, the antibacterial activity evaluations used samples of constant surface area in contact with the liquid medium.
Comment 8: This conclusion is not clear. After a careful statistical analysis, it must be clearly stated that MIL-100(Fe) show superior antibacterial activity compared to the corresponding samples containing TTO that show none (or little) activity against staphylococcus a.
Author’s reply: The Conclusions section was revised as instructed by Reviewer.

Round 2
Reviewer 1 Report
The manuscript entitled “MIL-100(Fe)-based composite films for food packaging”, submitted for evaluation to Nanomaterials, presents the proposal of complex biomaterial designed for food packaging, as an alternative for hardly biodegradable e.g. polyethylene foils.. In the context of the overwhelming wastes problem, I consider the topic of the work to be important and timely.
The manuscript was corrected and supplemented according to suggestions; however, Figure 5 still lacks statistical analysis of presented data. Please provide statistical significance values and point out which data are statistically different.
Author Response
Reviewer 1
The manuscript entitled “MIL-100(Fe)-based composite films for food packaging”, submitted for evaluation to Nanomaterials, presents the proposal of complex biomaterial designed for food packaging, as an alternative for hardly biodegradable e.g. polyethylene foils.. In the context of the overwhelming wastes problem, I consider the topic of the work to be important and timely.
The manuscript was corrected and supplemented according to suggestions; however, Figure 5 still lacks statistical analysis of presented data. Please provide statistical significance values and point out which data are statistically different.
Author’s reply: We have added two additional Tables with the data, requested by Referee.
Reviewer 3 Report
I recommend this paper for publication in the present form.
Author Response
No responce necessary.